



# Assimilation of synthetic radar backscatters at Ku-band improves SWE estimates

Nicolas R. Leroux[1], Vincent Vionnet[1], Courtney Bayer[1], Julien Meloche[2], Arlan Dirkson[1], Franck Lespinas[1], Mark Buehner[1], Marco Carrera[1], Benoit Montpetit[2], Bernard Bilodeau[1], Maria Abrahamowicz[1], and Chris Derksen[2]

[1]Meteorological Research Division, Environment and Climate Change Canada, Quebec, Canada
[2]Climate Research Division, Environment and Climate Change Canada, Ontario, Canada

**Correspondence:** Nicolas R. Leroux (nicolas.leroux@ec.gc.ca)

**Abstract.** In cold regions, snow serves as the primary water source for downstream rivers and lakes. Accurate gridded snow water equivalent (SWE) estimation is hindered by the sparse ground observation network and the low resolution of satellite passive microwave products. To address this, Environment and Climate change Canada (ECCC), the Canadian Space Agency (CSA), and Natural Resources Canada (NRCan) are developing the Terrestrial Snow Mass Mission (TSMM), a dual Ku-band

satellite mission designed to measure backscatter at 13.5 GHz and 17.25 GHz across the Northern Hemisphere at a 500-m spatial resolution with a weekly temporal resolution. This study assesses the feasibility of assimilating Ku-band backscatter to enhance SWE estimates in a synthetic experiment. We used the Soil-Vegetation-Snow version 2 (SVS2) land surface model, which incorporates the snowpack model Crocus, coupled with the Snow Microwave Radiative Transfer model (SMRT). Observations extracted at weekly intervals from synthetic truths (SWE and backscatter) were assimilated with a particle filter in

point-scale at three sites spanning different Canadian climates (Arctic, humid continental, Alpine) over three winter seasons. Meteorological forcing derived from the high-resolution Canadian meteorological model was perturbed to generate ensembles of snow simulations for assimilation. Results indicate that assimilating backscatter observations reduced the mean continuous ranked probability score (CRPS) of SWE estimates by up to 32 % at the Arctic and humid continental climate sites compared to the open-loop ensemble, performing similarly to the assimilation of SWE with an observation error larger than 20 %. As-

similating backscatter observations at the Alpine site only improved the SWE estimates by 5 % as backscatter signals seemed to lose sensitivity to SWE values greater than $\tilde{3}00\,\mathrm{kg\,m^{-2}}$ in our experimental setup. Assimilating backscatter and SWE observations also improved the estimations of vertical profiles of snow density and specific surface area. These findings demonstrate the potential of direct assimilation of Ku-band backscatter to enhance both estimates of SWE and snowpack properties.

## 1 Introduction

Gridded snow water equivalent (SWE) estimates are an essential component for water and food security, ecosystem sustainability, and predicting flood and drought risks in cold regions (Pomeroy et al., 2016; Wagner et al., 2017; Sturm et al., 2017). As the climate is changing, snow is being altered both spatially and temporally, which is likely to increase avalanche risk and flooding in both the short and long term (Haeberli and Whiteman, 2015; Hamlet and Lettenmaier, 2007; Beniston, 2012; Wagner et al.,



2017). Distributed SWE estimates from snowpack models help understand the spatial distribution of the snow properties over
a domain compared to sparse observations, and can be used to initialize land surface and hydrological forecasts (Orio-Alonso
et al., 2023; Garnaud et al., 2021). However, SWE estimates from snow models are subject to uncertainties linked to differ-
ent parameterization of snow processes (Lafaysse et al., 2017) and uncertainties in the meteorological forcings, in particular
in the precipitation amount and phase (Günther et al., 2019; Raleigh et al., 2015; Vionnet et al., 2022; Leroux et al., 2023).
There is a critical need for improved estimates of SWE at high temporal and spatial resolutions to advance reanalysis products,
operational models, and climate projections (Girotto et al., 2020).

Data assimilation is a commonly used method to decrease uncertainty in snow predictions by combining observations and
model estimates. Observations used for snow data assimilation can originate from in-situ manual snow observations (Orio-
Alonso et al., 2023; Magnusson et al., 2014; Oberrauch et al., 2024; Piazzi et al., 2018; Slater and Clark, 2006; Brasnett,
1999), remote sensing (Cluzet et al., 2020; De Lannoy et al., 2012; Durand et al., 2009; Larue et al., 2018a; Aalstad et al.,
2018; Li et al., 2017; Su et al., 2008, 2010; Lahmers et al., 2022), or retrieved SWE products (Shrestha and Barros, 2025b;
De Lannoy et al., 2012). Remote sensing observations for snow data assimilation predominantly utilize passive microwave
sensors (Durand et al., 2009; Larue et al., 2018a; Lemmetyinen et al., 2018; Li et al., 2017; Dumont et al., 2012) or snow
cover fraction and albedo products derived from optical sensors (De Lannoy et al., 2012; Garnaud et al., 2021; Aalstad et al.,
2018; Su et al., 2008, 2010). Additional satellite-derived and airborn remote sensing datasets have been explored for snow
data assimilation, including terrestrial water storage (TWS) information from the Gravity Recovery and Climate Experiment
(GRACE) satellites (Su et al., 2010), snow depth from Pleiades imagery (Marti et al., 2016; Shaw et al., 2020), snow depth
retrieved from ICESat-2 (Mazzolini et al., 2024), and SWE estimates from the NASA Airborne Snow Observatory (Lahmers
et al., 2022).

Data assimilation of both in-situ and satellite observations can improve SWE estimates, though each observation type has dif-
ferent advantages and limitations. In-situ snow depth measurements provide better SWE estimates compared to satellite optical
data when assimilated (e.g. Charrois et al., 2016), but the sparse distribution of in-situ measurements limits spatial representa-
tiveness (e.g. Alonso-González et al., 2023) and poses challenges when propagating information to unobserved regions (Pflug
et al., 2024; Cluzet et al., 2022). Satellite observations offer broader spatial coverage but present other limitations. Passive
microwave data provide global coverage with 40-year time series (e.g. Parkinson, 2022), but their coarse spatial resolution (e.g
Larue et al., 2018b) makes derived snow products unreliable in mountainous terrain (Luojus et al., 2021). Active microwave
sensors can retrieve SWE at higher spatial resolution (<500m) and for deeper snowpacks (Tsang et al., 2022), providing an
alternative to passive systems. Optical sensors face limitations from cloud cover, which affects data availability in the visible
and near-infrared ranges (De Lannoy et al., 2012). Both optical and microwave remote sensing data have higher uncertainties
for wet snow conditions and forested landscapes, which can limit accurate snow-covered area classification (Gascoin et al.,
2024; Muhuri et al., 2021).

To address the need for more accurate, high spatial resolution SWE estimates from remote sensing data, Environment and
Climate change Canada (ECCC), in partnership with the Canadian Space Agency (CSA) and Natural Resources Canada (NR-
Can), is developing the Terrestrial Snow Mass Mission (TSMM) (Derksen et al., 2021). This mission will be the only active





synthetic aperture radar (SAR) mission dedicated to snow, aiming to launch a dual-frequency Ku-band satellite (13.5 GHz

and 17.25 GHz) with a weekly revisit frequency and 500-meter resolution. TSMM is designed to complement existing passive

microwave observations, thereby enhancing SWE retrieval capabilities (Lemmetyinen et al., 2018). Recent studies highlight

the significance of Ku-band response to variations in SWE and snow microstructure, which are critical factors for SWE re-

trieval approaches utilizing Ku-band backscatter measurements (Lemmetyinen et al., 2018; Montpetit et al., 2025; Tsang et al.,

2022). Previous research exploring the potential of assimilating retrieved SWE from Ku-band and/or X-band volume-scattering

data has shown promise in improving snow profile properties (Phan et al., 2014) and SWE estimates (Cho et al., 2022; Pflug

et al., 2024; Shrestha and Barros, 2025b; Garnaud et al., 2019). Specifically, Cho et al. (2022) demonstrated that improve-

ments in SWE prediction root-mean-square error (RMSE) were more pronounced during the melting season than during the

accumulation period in the mountainous regions of Colorado. Their work also indicated that lower assimilation RMSE were

expected if retrieval algorithms are able to retrieve a wider range of SWE values, while higher tree cover fractions negatively

impacted assimilation performance. To mitigate this, Pflug et al. (2024) proposed a method to estimate forest SWE from adja-

cent forest-free areas to improve SWE prediction in forested environments via data assimilation. More recently, Shrestha and

Barros (2025b) showed that assimilating retrieved SWE from X- and Ku-band backscatter not only reduced snow depth and

SWE model biases and errors in vertical snow density profiles, but also improved the forward simulation of volume backscatter.

They also noted that larger observation errors in SWE retrievals led to increased RMSE in SWE prediction.

Durand et al. (2009) demonstrated that directly assimilating passive microwave radiance observations provides more accurate

snow depth predictions compared to assimilating retrieved SWE. Similarly, in soil moisture research, assimilating brightness

temperature measurements from passive microwave satellite observations is often preferred over retrieved soil moisture data

to enhance estimation accuracy (e.g., De Lannoy and Reichle, 2016; Carrera et al., 2019). A significant limitation of retrieved

products (Level 2 data) is their delayed availability, often hours after the original observations (Level 1 data), which poses

challenges for near-real-time operational assimilation systems. To date, snow data assimilation studies have not yet explored

the potential of assimilating Ku-band backscatter observations to improve SWE estimates. This study aims to address this gap

in the literature.

Point-scale synthetic experiments are a common approach to evaluate the feasibility of assimilation schemes, utilizing syn-

thetic observations derived from model runs (Charrois et al., 2016; Durand and Margulis, 2006; Larue et al., 2018a; Revuelto

et al., 2021). The objective of this work is to demonstrate the potential of direct Ku-band backscatter assimilation to enhance

SWE estimation through a synthetic point-scale experiment. This is made possible by the recent development of a new land

surface scheme (Soil-Vegetation-Snow version 2, SVS2) at ECCC that includes the multi-layered snow model Crocus (Vionnet

et al., 2025). SVS2 can provide the necessary snow inputs to a forward radiative transfer model, such as the Snow Microwave

Radiative Transfer model (SMRT) (Picard et al., 2018) to estimate backscatter at Ku-band. SVS2/Crocus coupled with SMRT

were included in the Multiple Snow Data Assimilation System (MuSA) platform (Alonso-González et al., 2022) to develop the

synthetic data assimilation experiment detailed in this paper. Due to the strong non-linearity of multi-layered snow models, the

particle filter is used for the data assimilation as in Cluzet et al. (2021); Charrois et al. (2016); Larue et al. (2018a); Revuelto

et al. (2021). In this study, SWE estimates derived from assimilating backscatter observations are compared against SWE es-





timates obtained by assimilating SWE observations with varying levels of uncertainty. Section 2 presents the study sites and

data used, the models applied, and the synthetic experiment design. The synthetic experiment results on bulk snow properties prediction (SWE and snow depth) and vertical snow properties (density and snow specific area, SSA) are presented in Sect. 3 and discussed in Sect. 4.

## 2   Design of the synthetic experiments

### 2.1   Study Sites

The synthetic experiment is run at three study sites that span different Canadian climates: Trail Valley Creek (TVC) is an Arctic site in the Northwest Territories (68.74°, -133.5°, elevation of 91 m above see level, asl), Rogers Pass is an Alpine site in British Columbia (51.23°, -117.71°, elevation of 1905 m asl), and Powassan is an agricultural site, in a humid continental climate in Ontario (46.08°, -79.36°, elevation of 256 m asl) (Fig. 1). These sites were chosen because field experiments for science readiness activities of TSMM were conducted there, gathering in-situ snow observations (snow pits and snow courses),

airborne and ground-based Ku-band backscatter, and meteorological observations between 2018 and 2024 (Montpetit et al.; Kelly et al., 2024; Madore et al., 2023). The study period includes three consecutive winters, from September 2020 to August 2023. Table 1 summarizes the total snowfall amounts and mean air temperature between September 1 and June 30 at each site for each winter season extracted from the High Resolution Deterministic Prediction System (HRDPS, Milbrandt et al. (2016)) and the Canadian Precipitation Analysis System (CaPA) (Khedhaouiria et al., 2020; Lespinas et al., 2015) using Wang

et al. (2019a) for the precipitation phase partitioning (Sect. 2.3.1). TVC receives less snow than the other sites, while Rogers Pass received significantly more snow. Powassan is the warmest site with mean air temperatures above freezing and TVC the coldest site. Meteorological data, including air temperature, relative humidity, and wind speed, were measured at Powassan during the 2022-2023 winter season (Kelly et al., 2024), and were used to inform the forcing perturbations needed for the ensemble generation (Sect. 2.3.1).





**Table 1.** Total snowfall (water equivalent) and mean air temperature (Sept 1 - June 30) at each study site

| Site | Winter Season | Total Snowfall (mm w.e.) | Mean Air Temperature (°C) |
|------|---------------|--------------------------|---------------------------|
| | 2020-2021 | 194 | 3.6 |
| Powassan | 2021-2022 | 204 | 3.0 |
| | 2022-2023 | 303 | 4.2 |
| | 2020-2021 | 52 | -12.0 |
| TVC | 2021-2022 | 129 | -12.2 |
| | 2022-2023 | 128 | -11.2 |
| | 2020-2021 | 1058 | -3.5 |
| Rogers Pass | 2021-2022 | 1419 | -4.9 |
| | 2022-2023 | 784 | -3.4 |

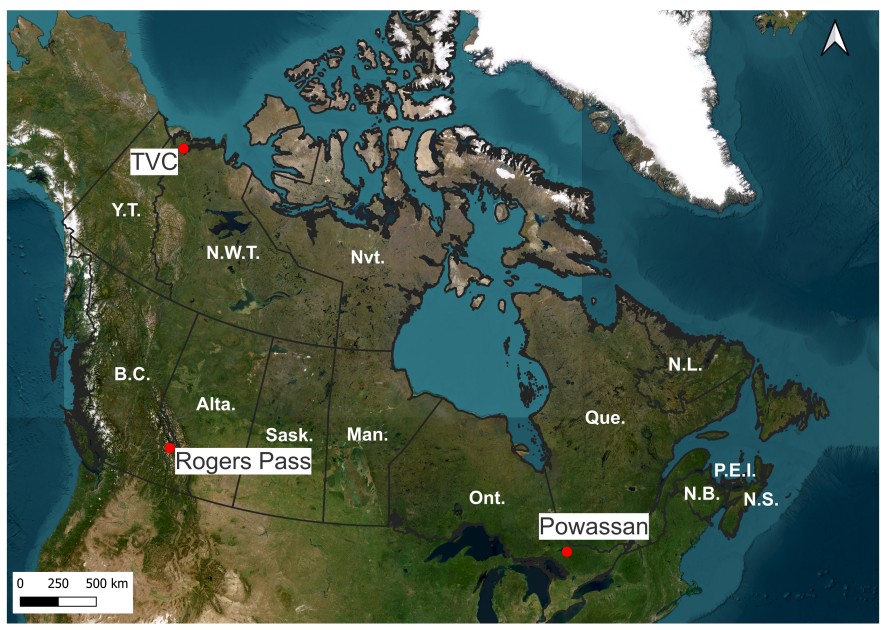

**Figure 1.** Map showing the locations of the three sites: Trail Valley Creek (TVC), Rogers Pass, and Powassan. Basemap: ESRI world imagery.

## 2.2 SVS2/Crocus and SMRT

The land surface scheme SVS2 developed at ECCC (Vionnet et al., 2025) contains the detailed snowpack model Crocus (Vionnet et al., 2012; Lafaysse et al., 2017, 2025). Crocus is a 1D snowpack model that simulates the seasonal evolution of the physical properties of the snowpack and its vertical layering. For each snow layer, Crocus simulates the evolution of the thickness, density, liquid water content, temperature, age, and snow microstructure represented by the snow specific surface



area and the snow grain sphericity (Brun et al., 1992; Vionnet et al., 2012; Carmagnola et al., 2014). Crocus was initially developed to simulate the properties of alpine snow in the context of avalanche hazard forecasting (e.g. Durand et al., 1993). Recently, Woolley et al. (2024) proposed an Arctic configuration of SVS2/Crocus that improves the simulations of Arctic snowpack properties through better representation of wind-packing and inclusion of the effect of basal vegetation on snow compaction. This Arctic version of Crocus was used at TVC and the default version of Crocus was used at Powassan and

Rogers Pass (see Table A1 for the Crocus parameterizations). Montpetit et al. (2025) used the Arctic version of Crocus to successfully retrieve SWE at TVC from Ku-band SAR measurement. In our study, a maximum of 20 snow layers was specified to simulate the evolution of the snowpack properties with SVS2/Crocus.

SMRT is a snow radiative transfer model (Picard et al., 2018) that was coupled with SVS2 (Meloche et al., 2025). SMRT was used to compute backscatter signal ($\sigma$) at the two TSMM frequencies, 13.5 GHz and 17.25 GHz in VV polarization,

from simulated snow layered properties (density, thickness, SSA, and temperature) from SVS2/Crocus. The DORT solver was used with the Improved Born Approximation and an exponential microstructure. An incidence angle of 35° was assumed. The simulated soil temperature and water content from SVS2 in the upper 5 cm of the soil were used to calculate the soil permittivity using the Mironov model (Mironov et al., 2019).

## 2.3   Assimilation design

SVS2, coupled with SMRT, was included in the version 2.0 of the MuSA (Alonso-González et al., 2022). MuSA is a Python toolbox that can generate ensembles of snow simulations and offers a wide selection of assimilation methods. In this study, the particle filter method was used as it is suited for non-linear models, such as multilayered snow models. The particle filter has been used extensively with complex snow models (Cluzet et al., 2021; Magnusson et al., 2017; Revuelto et al., 2021; Larue et al., 2018a; Charrois et al., 2016).

### 2.3.1   Generation of the Ensemble

Because uncertainties in the snow simulations were assumed to originate from uncertainties in the meteorological forcing (Raleigh et al., 2015; Günther et al., 2019), the snow ensemble were generated by only perturbing the meteorological forcing, Meteorological forcing for SVS2/Crocus was obtained at each site from the HRDPS at 2.5 km grid spacing. These forcing included air temperature, specific humidity, wind speed, surface pressure, and incoming longwave and shortwave radiation.

Successive short-term HRDPS forecasts (7-12 hr lead time) were combined to generate continuous hourly meteorological forcing. Air temperature and humidity were at 2-m agl and wind speed was at 10-m agl. The precipitation amount was taken from the CaPA 2.5-km (Khedhaouiria et al., 2020; Lespinas et al., 2015). Once the meteorological forcings were perturbed, the phase of the precipitation was determined using the approach of Wang et al. (2019b) that relies on the near-surface wet-bulb temperature.

The time evolution of the perturbations applied to the meteorological forcing followed a first-order auto-regressive model describing the time evolution of an error as in Magnusson et al. (2017):



$$q_k = \alpha q_{k-1} + \sqrt{1-\alpha^2}\,\omega_k \tag{1}$$

where $q_\mathrm{k}$ is the error at time $k$, $\omega_\mathrm{k}$ is a white noise with mean of 0 and standard deviation of 1 that changes with $k$, and $\alpha$ is a function of the decorrelation time length $\tau$:

$$\alpha = 1 - \frac{\Delta t}{\tau} \tag{2}$$

with $\Delta t$ being the model time step.

   Random perturbations were applied to the meteorological inputs either as additive or multiplicative perturbations. The additive perturbations were drawn from a normal distribution and were applied to the air temperature and incoming longwave radiation forcing while the multiplicative perturbations were drawn from a log-normal distribution and were used for the precipitation, wind speed, and shortwave radiation forcing.

$$n_k = \mu_f + q_k SD_f \qquad\qquad \text{if additive perturbation} \tag{3}$$

$$n_k = \exp(\mu_f + q_k SD_f) \qquad\qquad \text{if multiplicative perturbation} \tag{4}$$

where $n_k$ is the perturbation applied to the meteorological forcing at time $k$. For additive perturbations, $\mu_f$ and $SD_f$ are the mean and standard deviation of the normal distribution, respectively. For multiplicative perturbations, they are the parameters of the underlying normal distribution in the log-normal formulation. For the normal distributions, we assumed $\mu_f$ equal to 0 and $SD_f$ is calculated from the differences between HRDPS model predictions and actual meteorological observations at Powassan during 2022-2023 (Sect. 2.1). For the log-normal distributions (Eq. 4), we calculated $SD_f$ as the standard deviation of the logarithm of the ratio between HRDPS predictions and observations, and adjusted $\mu_f$ so that the mean of $n_k$ equaled 1. This approach made it possible to base the uncertainty of our perturbations on discrepancies in the HRDPS forcing data. The variable $\alpha$ in Eq. 1 was determined by calculating the autocorrelation of the residual of each variable between the HRDPS and the observations with a lag of 1. For simplicity, these discrepancies between observations and HRDPS were assumed to hold true at the two other sites. Table 2 summarizes the parameters used to generate the perturbations ($\tau$ was determined from the calculated $\alpha$ values and Eq. 1). As direct longwave radiation measurements were not available for determining its perturbation, a linear regression was determined using the HRDPS forcing data, modelling changes in longwave radiation as a function of changes in air temperature (Table 2). This method propagates the air temperature uncertainty to longwave radiation based on their HRDPS-derived correlation. Based on this perturbation strategy, a total of 100 members were generated for each assimilation experiment as it was found suitable for snow assimilation with the particle filter (Piazzi et al., 2018).




**Table 2.** Parameters of the perturbation applied to the meteorological forcing to generate the ensemble

| Variable | Distribution | $\mu_f$ | $SD_f$ | $\tau$ (h) |
|---|---|---|---|---|
| Air temperature ($^o$C) | normal | 0 | 1.46 | 10.3 |
| Precipitation (km m$^{-2}$ h$^{-1}$) | Log-normal | -0.22 | 0.67 | 24 |
| Wind speed (m s$^{-1}$) | Log-normal | -0.05 | 0.31 | 2.6 |
| Shortwave radiation (W m$^{-2}$) | Log-normal | -0.005 | 0.01 | 3 |
| Longwave radiation (W m$^{-2}$) | Linear regression* | - | - | - |

*The linear regression between the longwave radiation perturbation ($\epsilon_{k,LW}$) and air temperature perturbation ($\epsilon_{k,T_a}$) is $\epsilon_{k,LW} = 3.7\epsilon_{k,T_a}$ with $\epsilon_{k,T_a}$ in K.

### 2.3.2 Data Assimilation Experiments

This study focuses on idealized experiments, in which synthetic observations were generated from reference model runs and subsequently assimilated (e.g., Revuelto et al., 2021; Durand and Margulis, 2006; Larue et al., 2018b). For each winter season and site, we generated 10 reference runs of SVS2/SMRT using perturbed meteorological inputs (Sect. 2.3.1). These runs were used as synthetic true states providing reference snow states and backscatter values. We used multiple reference runs to evaluate data assimilation performance across different snowpack conditions within each winter season, following the approach of Revuelto et al. (2021). Because the main source of uncertainties is assumed to originate from the meteorological forcing (Sect. 2.3.1), the model parameterization between these reference runs and the ensemble members was identical. SWE and backscatter values were extracted from the reference runs at weekly intervals on Mondays at 1200 UTC, which is the expected revisit period of TSMM (but not particularly the expected date of observation at the sites and the time was chosen to avoid wet snowpack observations potentially occurring later in the day) so long as the following criteria are met: 1) bulk liquid water content within the snowpack below 1% of mass as TSMM measurements for wet snowpacks would be discarded, and 2) backscatter values at 13.5 GHz and 17.25 GHz above -24 dB and -26 dB, respectively, which correspond to the lowest snow backscatter observations at these frequencies from a tower-based radar system done by Derksen et al. (2021).

A series of assimilation experiments were conducted to evaluate the impact of different observation types and their associated uncertainties. The synthetic observations were generated by adding random noise to the values extracted from the reference runs, which are considered to be the true snowpack states. The noise was drawn from a normal distribution with a standard deviation equal to the observation uncertainty. The following observations were assimilated: 1) SWE with uncertainty ranging from 5 % - the best expected accuracy of manual measurements (Beaudoin-Galaise and Jutras, 2021)) - to 30 % - corresponding to typical uncertainties in radar-based SWE retrievals (e.g. Montpetit et al., 2025; Pomerleau et al., 2020; Cho et al., 2022)); 2) backscatter at 13.5 GHz with a 1 dB uncertainty; 3) backscatter at 17.25 GHz with a 1 dB uncertainty; 4) both 13.5 GHz and 17.25 GHz backscatter simultaneously, with a 1 dB uncertainty for each frequency; and 5) the backscatter difference between 13.5 GHz and 17.25 GHz, with a 1.4 dB uncertainty, as sometimes employed in brightness temperature assimilation to enhance snow information extraction (e.g Larue et al., 2018b). The assumed 1 dB uncertainty reflects the target measurement accuracy





of individual TSMM observations. However, this does not include additional errors that would arise in practice, such as spatial mismatch between observation and model grid (representativeness error), inaccuracies in converting model states to backscatter (observation operator errors), or other systematic biases.

Following each assimilation step (i.e. observation time step), a new 100-member ensemble was generated from the assimilated particles using the particle filter's resampling process (Sect. 2.3.3). The forcing perturbations for the new 100 members were generated by drawing new samples from the noise distributions (Sect. 2.3.1).

     The 10 randomly chosen reference runs presented different snowpack conditions that can impact the assimilation experiments (Fig. 2). By construction of the synthetic experiment, the reference runs were mostly within the generated 100-member

ensembles. The snowpack was shallowest in TVC, deeper in Powassan, while the Alpine snowpack in Rogers Pass reached depths exceeding 3 m. The backscatter at 17.25 GHz was slightly higher than at 13.5 GHz. While the backscatter at Powassan and TVC increased with SWE, they rapidly reached a plateau at Rogers Pass early in the season.





**Figure 2.** Spread of the open loop (OL) ensemble composed of a 100 members (between 5th and 95th percentiles) and the 10 reference runs at Powassan (a, d, g, j), TVC (b, e, h, k), and Rogers Pass (c, f, i, l) for the backscatter at 13.5 GHz (a, b, c), for the backscatter at 17.25 GHz (d, e, f), for snow height (g, h, i), and for SWE (j, k, l) for the winter 2020-2021. The same figures for the other two winter seasons can be found in the supplementary material (Fig. S1 and S2).

### 2.3.3 The Particle Filter

The particle filter with Sequential Importance Resampling (PF-SIR, Gordon et al. (1993)) was used for the assimilation of
different snow variables described in Sect. 2.3.2. In the particle filter, the prior distribution (also referred to as background) of the model states $x_t$ at time $t$, $p(x_t|y_{1:t-1})$, is compared to the observations at time $t$, $y_t$, to estimate a posterior distribution, $p(x_t|y_t)$, from calculated weights, $w$, between the prior distribution and the observations:

$$p(x_t|y_t) \propto w\,p(x_t|y_{1:t-1}) \tag{5}$$





The prior distribution is composed of an ensemble of model states (particles). During the first step of the PF-SIR, the prior

particles are weighted based on their distance to the observation. The weight for each ensemble member $i$ is calculated as:

$$w_\mathrm{i} = \frac{\exp(-\frac{1}{2}[\boldsymbol{y} - \hat{\boldsymbol{y_\mathrm{i}}}]\boldsymbol{R}^{-1}[\boldsymbol{y} - \hat{\boldsymbol{y_\mathrm{i}}}])}{\sum_{\mathrm{j}=1}^{N_\mathrm{e}} \exp(-\frac{1}{2}[\boldsymbol{y} - \hat{\boldsymbol{y_\mathrm{j}}}]\boldsymbol{R}^{-1}[\boldsymbol{y} - \hat{\boldsymbol{y_\mathrm{j}}}])} \tag{6}$$

where $\boldsymbol{R}$ is the error covariance matrix that accounts for the uncertainty in the observations, $N_\mathrm{e}$ is the total number of members, and $\hat{y}_\mathrm{i} = H(x_\mathrm{i})$ with $H(\cdot)$ is the observation operator which maps from the model state to the observation space. During the second step of the PF-SIR, a resampling of the particles is conducted to select the particles with the highest weights. In our

study, a systematic resampling is used and all selected particles are then duplicated to generate a new ensemble composed of $N_e$ members (100 members in this study). The particle filter is known to suffer from degeneracy when all ensemble members collapse to a few particles (e.g., Moradkhani et al., 2012). To monitor degeneracy, the effective sample size ($N_{eff}$) was calculated after the calculation of the weights:

$$N_{eff} = \frac{1}{\sum_{i=1}^{N_e} w_i^2} \tag{7}$$

In our study, we considered that degeneracy happened when $N_{eff}/N_e$ is below 7 % as in Larue et al. (2018a); Cluzet et al. (2021).

### 2.3.4 Metrics for Evaluation

The RMSE and the continuous ranked probability score (CRPS) were computed to evaluate the performance of the different assimilation experiments as follow:

$$\text{RMSE} = \sqrt{\frac{(\bar{\mathbf{x}} - \mathbf{y})^2}{M}} \tag{8}$$


where $\bar{\mathbf{x}}$ is the mean of the ensemble members at all the observation times, $M$ is the number of observations, and $\mathbf{y}$ are the values of the reference runs at the observations times.

$$\text{CRPS}_t = \int R(F_t(x) - O_t(x))^2 dx \tag{9}$$

where $\text{CRPS}_t$ is the CRPS at the observation time $t$, $F_\mathrm{t}(x)$ is the cumulative distribution function of the analysis predictions at

time $t$ and $O_\mathrm{t}(x)$ is the corresponding cumulative distribution function of the reference run value at time $t$. $\text{CRPS}_t$ were then averaged over time for each assimilation experiment to obtain a mean CRPS.

These scores were calculated for the open loop ensemble, the background particles, and the assimilated particles (the analysis). The scores obtained from the open loop served as a baseline for assessing improvements in snow predictions (SWE and snow depth) achieved through assimilation. Comparing the scores between the analysis and the background particles high-

lighted the performance of the assimilation at each assimilation step. The RMSE was calculated for the mean of the ensemble





and the observations to evaluate the performance of the ensemble means, while the CRPS was applied to all ensemble members. A single RMSE and CRPS value were computed for each distinct experimental configuration, which comprised a unique reference run and year.

The RMSE, and CRPS of the assimilation results were normalized against those of the open-loop or the background particles. The normalized CRPS against the open loop ($CRPS_{norm, OL}$) was calculated as:

$$CRPS_{norm, OL} = 100x(CRPS_{OL} - CRPS_{exp})/CRPS_{OL} \tag{10}$$

where $CRPS_{OL}$ is the CRPS of the open loop and $CRPS_{exp}$ is the CRPS of the assimilation for a same reference run and winter. Negative values of $CRPS_{norm, OL}$ meant that the assimilation performed worse than the open-loop and positive values showed an improvement of the assimilation over the open-loop. Similarly, we normalized CRPS scores against the background particles ($CRPS_{back}$).

We evaluated the simulated vertical profiles of snow density and SSA, both with and without assimilation, against those from the reference runs. First, the height of all vertical profiles was normalized between 0 and 1 to allow a direct comparison as in Woolley et al. (2024). We then divided this normalized height into equally spaced layers, each with a thickness of 0.005 (Viallon-Galinier et al., 2020). Subsequently, the vertical density and SSA values were compared at each of these 0.005 layers, and the RMSE and CRPS for each layer were averaged per profile. This analysis aimed at quantifying improvements in estimated snowpack vertical properties via the assimilation method compared to the open-loop ensemble members.

## 3 Results

In this section, we first present the results of the assimilation experiments on bulk SWE and snow depth estimates. These results are compared across the different types of assimilated observations. Finally, we examine how well each observation type improves vertical snow profile estimation (density and SSA) beyond the open loop baseline.

### 3.1 Estimation of Snow Bulk Properties

Figure 3 presents seasonal evolution of SWE at the three sites in 2022-2023 obtained from three assimilation experiments, using a single reference run across all three sites. These experiments include: assimilation of 13.5 GHz backscatter observations, assimilation of 17.25 GHz backscatter observations, and assimilation of SWE observations with a 10 % uncertainty. Generally, the assimilation experiments showed improved SWE predictions by reducing the spread of the ensemble compared to the open loops. At all sites, backscatter assimilation revealed a spread increasing during the accumulation period and diminishing during the melt season. At Powassan and Rogers Pass, SWE predictions after assimilating backscatter observations at 17.25 GHz presented a slightly smaller spread than those from backscatter at 13.5 GHz, although this narrower spread did not always encompass the reference run. The assimilation of backscatter at Rogers Pass barely improved the SWE estimate compared to the open loop, despite this site having the highest number of weekly observations throughout the winter. On the other hand, assimilating synthetic SWE observations with an error of 10 % provided an analysis more centered around the reference run




with a much reduced spread than when assimilating either backscatter. Overall, assimilating SWE observations provided SWE estimates with a narrower spread around the reference run compared to assimilating backscatter observations.

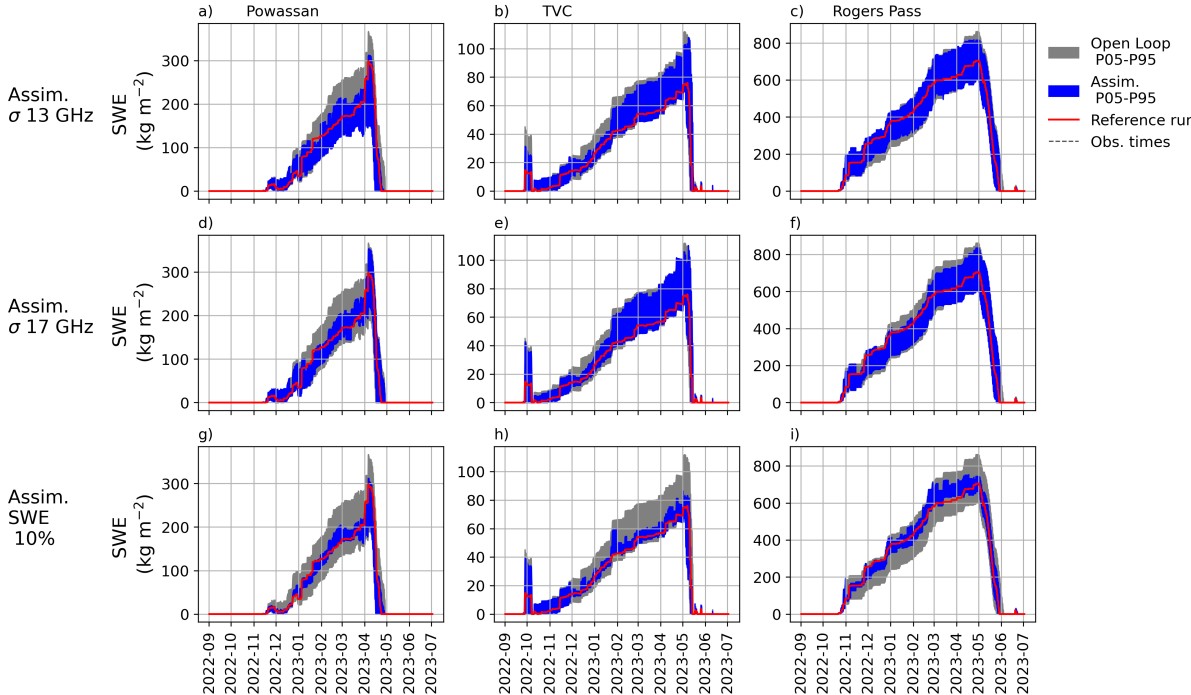

**Figure 3.** Results of the assimilation of the reference run #1 at Powassan (a,d,g) for the 2022-2023 snow season, TVC (b,e,h), and Rogers Pass (c,f,i). (a,b,c) are the results of assimilating backscatter observations at 13.5 GHz, (d,e,g) for the assimilation of backscatter observations at 17.25 GHz, and (g,h,i) for the assimilation of SWE observations with an error of 10 %. The gray envelope shows the spread between the $5^{th}$ and $95^{th}$ percentiles of the open loop (OL) without assimilation, while the blue envelopes show the results of the assimilation ($5^{th}$ to $95^{th}$ percentiles) and the corresponding reference run is in red.

In most cases, assimilating the individual backscatter observations improved SWE estimates compared to the open loop (Fig.
4a). The TVC site consistently shows larger improvements with mean normalized CRPS values of SWE estimates exceeding 30 % compared to Powassan (~23 %) and Rogers Pass (between 0 % and 3 %). The assimilation of either backscatter provided similar results in terms of mean normalized CRPS, but the assimilation of the backscatter at 17.25 GHz slightly improved the estimates at Powassan (+1.5 % compared to 13.5 GHz for SWE estimates) but the assimilation of backscatter at 13.5 GHz worked better at Rogers Pass (+3 % compared to 17.25 GHz for SWE estimates) and at TVC (+1.5 % compared to 17.25 GHz
for SWE estimates). Very little improvements in SWE estimates were found over the open loop at Rogers Pass when assimilating the backscatter at 17.25 GHz. The assimilation of individual backscatter showed greater improvements for SWE predictions at Powassan and TVC than for snow depth predictions (Fig. 4b), with an increase of ~10 % and ~6 % of mean normalized CRPS, respectively. The normalized RMSE were qualitatively similar to the normalized CRPS (Fig. S3 of the supplementary



material), showing the positive impact of the assimilation of backscatter to improve the mean prediction accuracy of SWE and
snow depth estimates.

    Assimilating both frequencies demonstrated the highest gains in prediction accuracy at Powassan and TVC, outperforming the assimilation of individual frequencies with an increase in mean normalized CRPS of 10 % for SWE and snow depth estimates at Powassan and between 3 and 5 % at TVC (Fig. 4). This improvement was not systematic at Rogers Passs. In addition, the combined assimilation approach led to a reduction in the spread of normalized scores at both Powassan and TVC
(as shown by the size of the boxplots), indicating a more robust and less variable assimilation outcome for these locations. In contrast, Rogers Pass exhibited a wider spread of normalized scores under the dual-frequency assimilation, suggesting less consistent performance. Lastly, assimilating the difference between the frequencies showed slight improvements in normalized scores at Powassan and Rogers Pass for both SWE and snow depth estimates compared to assimilating individual frequencies (+2-4 % in mean normalized CRPS).

Assimilating SWE observations with an uncertainty of 5 % showed the best estimates of SWE and snow depth, with mean normalized CRPS values mostly above 75 % and 60 % for SWE and snow depth estimates, respectively. Improvements at TVC were greater than those of the other sites. As the uncertainty in SWE observation increases, the improvements of the assimilation compared to the open loop were reduced. Assimilating SWE with uncertainties equal to or below 20 % performed better than assimilating the backscatter observations, with improvements in mean normalized CRPS commonly above 45 % for
SWE estimates and 35 % for snow depth estimates. However, when assimilating SWE with an uncertainty of 30 %, the results were similar to assimilating the backscatter, except at Rogers Pass where it still performed better.



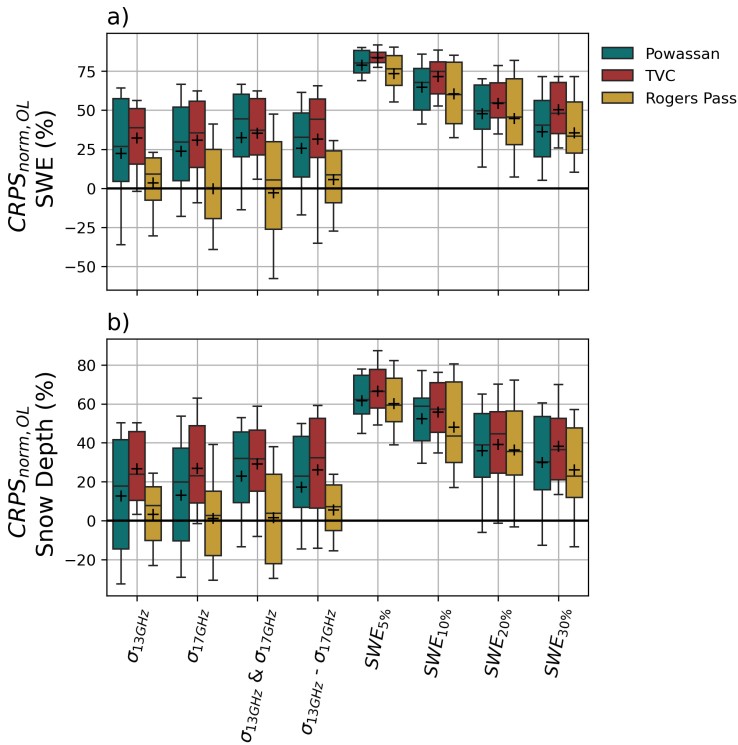

**Figure 4.** Normalized CRPS against the open loop at all the observation times over all the different runs for the three winter seasons for SWE prediction (a) and snow depth prediction (b) at the three sites based on different observations being assimilated. Box plots show median (center line), interquartile range (box), $10^{th}$–$90^{th}$ percentiles (whiskers), and mean (+). No outliers are shown for clarity.

To understand the effectiveness of the assimilation method on reducing forecast uncertainty at each assimilation time step, the CRPS of the analyses were compared to those of the background particles. The performance of the particle filter in narrowing the spread of the background ensemble is evaluated for different periods of the winter season (Fig. 5). At all sites, the
assimilation algorithm did not improve on the background particles in the middle of the winter when the SWE at each site was the highest, i.e. between January and February for Powassan and between January and April for TVC and Rogers Pass, which have longer winter periods than Powassan. The assimilation method performed best during the accumulation and melt periods when SWE values were the lowest. It is important to note that the number of observations during the melt period were low as only dry snow conditions were assimilated. In the middle of the winter season, the particle filter failed to reduce the spread of
background particles when assimilating the different backscatter observations. Assimilating both backscatters performed best at Powassan across the winter and at TVC early in the winter season. Assimilating the difference of backscatter outperformed the other assimilation of backscatter at Rogers Pass during the melting period, with mean normalized CRPS against the background particles of ~40 % for both SWE and snow depth estimates. The particle filter performed significantly better across the winter when assimilating the SWE observations, with greater results with lower SWE uncertainties, but once again, the im-
provements were lower in the middle of the season. The normalized CRPS against the open loop (Fig. S4 of the supplementary




material) did not show this dependency on seasonality, meaning that the SWE or snow depth analysis consistently improved upon the open loop across the season. These results suggest that assimilating observations early in the winter had the greatest impact, enhancing SWE and snow depth estimates for the rest of the season.

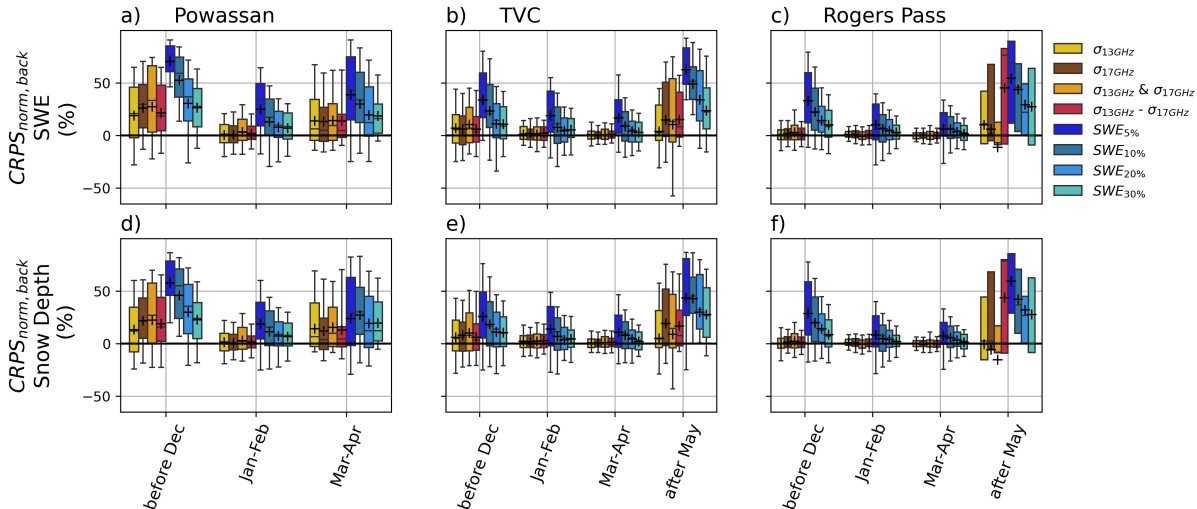

**Figure 5.** Normalized CRPS against the background particles for all the different runs and the three winter seasons for SWE prediction (a,b,c) and snow depth prediction (d,e,f) for Powassan (a,d), TVC (b,e), and Rogers Pass (c,f) based on the month of the observations. Box plots show median (center line), interquartile range (box), 10th–90th percentiles (whiskers), and mean (+). No outliers are shown for clarity.

## 3.2 Impact of Assimilation on the Vertical Snowpack Properties

The vertical profiles of density and SSA were compared for the different assimilation experiments with the corresponding reference run profiles. Figure 6 shows examples of density and SSA profiles at TVC on December 27, 2021, obtained with the experiments assimilating backscatter observations at 13.5 GHz and 17.25 GHz and SWE observations with an uncertainty of 10 %. In this example, assimilating SWE observations provided an estimate of the density and SSA profiles closer to the reference run than assimilating the individual backscatter observations, which only moderately improved the profiles of SSA

and density. When considering all the sites and all the reference runs over the three year period, the assimilation experiments produced simulated vertical density profiles that more closely matched the reference runs than those from the open loop simulations with positive median normalized CRPS values (Fig. 7). Assimilating individual backscatter at TVC improved the mean normalized CRPS of the snow density profiles compared to the open loop by $\tilde{8}$ % while they were around $\tilde{5}$ % at Powassan and 1 % at Rogers Pass. The distributions of normalized CRPS of the estimated profiles of SSA showed some improvements

with positive median values at TVC (up to 15 %) and almost no improvements at the other sites (median values close to 0), but presented mainly negative mean values that were impacted by under-performing outliers. Assimilating the different combinations of backscatter did not have significative improvements upon density and SSA profile estimates, with only slight improvements of density profiles when assimilating both backscatters at once at TVC and Rogers Pass. When assimilating



SWE observations, the vertical profiles of density and SSA were mostly improved compared to the open loop at the three sites, with larger improvements overall at Rogers Pass. As for bulk snow properties estimates, increasing the uncertainty in the SWE observations tended to decrease the estimates of the snow vertical profiles, with estimates similar to assimilating one or a combination of backscatter with SWE uncertainties of 30 %, except at Rogers Pass for density estimates.

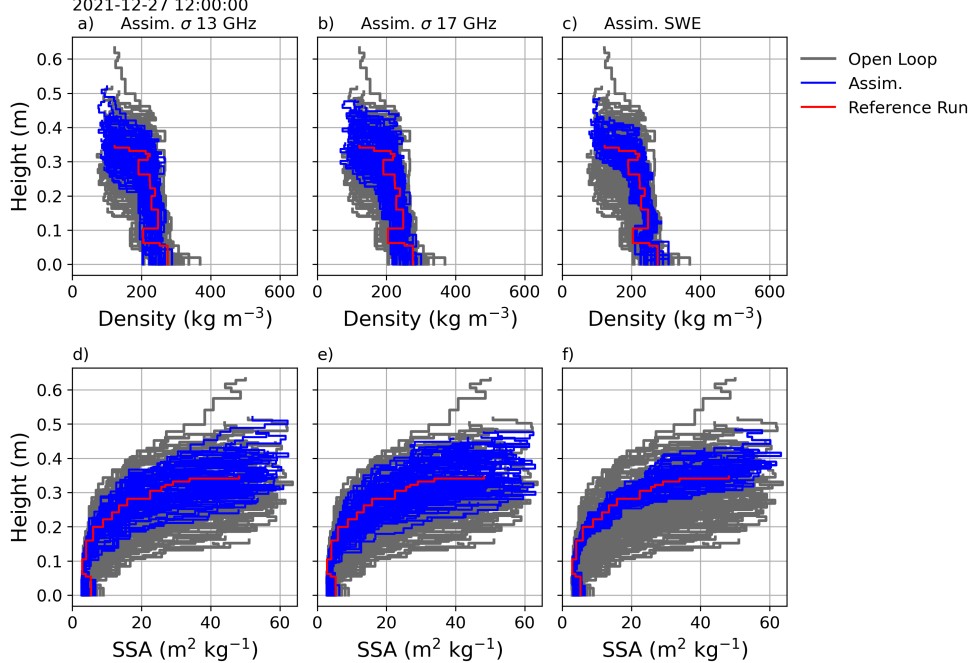

**Figure 6.** Results of the assimilation of the reference run #1 at TVC on the vertical profile of density (a,b,c) and for SSA (d,e,f) on December 27, 2021 1200 UTC. (a,d) are the results of assimilating $\sigma$ at 13.5 GHz, (b,e) of assimilating $\sigma$ at 17.25 GHz, and (c,f) of assimilation SWE with an error of 10 %.





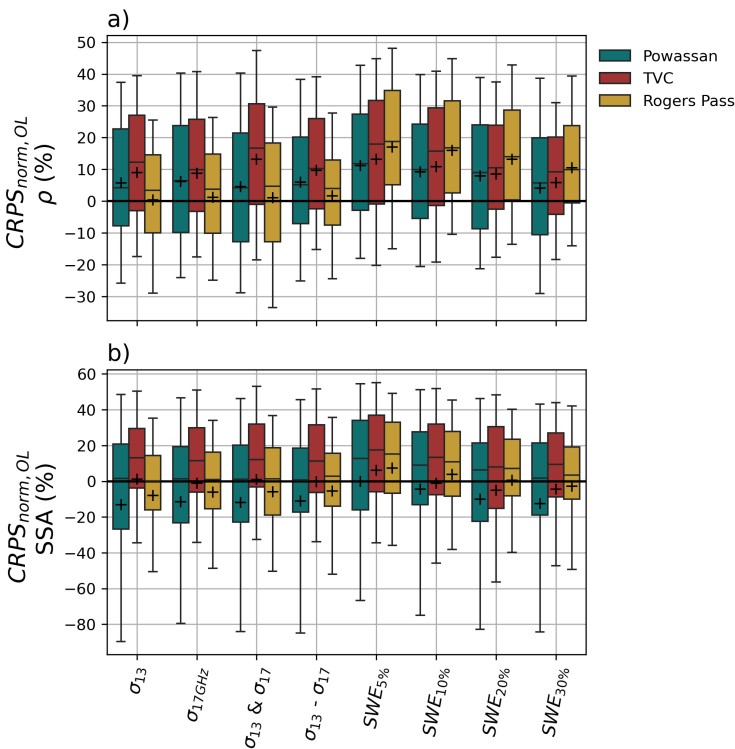

**Figure 7.** Normalized CRPS against the background particles for vertical density profiles prediction (a) and SSA vertical profiles (b) at the three sites (Powassan, TVC, and Rogers Pass) for the different assimilation experiments. Box plots show median (center line), interquartile range (box), 10th–90th percentiles (whiskers), and mean (+). No outliers are shown for clarity.

## 4 Discussion

To support the upcoming Terrestrial Snow Mass Mission (TSMM) (Derksen et al., 2021), this study demonstrates how assim-
ilating Ku-band backscatter observations at 13.5 GHz and 17.25 GHz through a particle filter can improve SWE and snow depth estimates. As a first step, we used a 1D point scale synthetic experiment with a novel snow data assimilation framework, SVS2 (Vionnet et al., 2025) coupled to the forward SMRT model (Picard et al., 2018) and the assimilation platform MuSA (Alonso-González et al., 2022). The particle filter proposed in MuSA was used in this study as this assimilation method has been extensively tested with multi-layered snowpack models (e.g. Charrois et al., 2016; Cluzet et al., 2021; Larue et al., 2018b;
Revuelto et al., 2021).

### 4.1 Backscatter Assimilation Improves Snow Estimates

The SWE estimates showed positive improvements after data assimilation of backscatter, with site-specific variations across sites representing different climate zones in Canada. At TVC, located in the Arctic, SWE estimates from individual backscatter



frequency assimilation demonstrated the most promising results, with mean CRPS improvements ranging between 30 % and
35 % over the open loop ensemble. In contrast, Powassan, situated in a humid continental climate, showed slightly lower
improvements, with improvements of $\tilde{2}3$ % over the open loop of mean CRPS. These differences can be attributed to several
key factors. First, the SVS2-Crocus configuration at TVC used in this study came from Woolley et al. (2024) as its best
represented the vertical profiles of density and SSA at the Arctic site, while the default SVS2-Crocus configurations were
specified at Powassan. In addition, TVC experiences colder winter conditions, providing between 20 and 30 valid observations
during the snow season. On the other hand, the snowpack at Powassan experiences more frequent melt and rain-on-snow events,
limiting observation availability to between 9 and 16 data points per winter. Consequently, the higher number of assimilated
observations at TVC may have contributed to the improved SWE and snow depth estimates. The winter conditions at Powassan
resulted in frequent melting and precipitation events producing complex vertical snow profiles, particularly with the formation
of ice lenses. While these features may not have impacted the assimilation results in this synthetic experiment, it is expected
that in real-world data assimilation, complex vertical snow layering due to melt events significantly impact microwave signals
and pose considerable challenges for radiative transfer model simulations (Bartsch et al., 2007; Dolant et al., 2018; Picard
et al., 2018; Montpetit et al., 2013).

Assimilation strategies combining the individual frequencies were tested as often done in snow assimilation studies (e.g.
Revuelto et al., 2021; Durand and Margulis, 2006) but yielded mixed results. At Powassan, combining both backscatter fre-
quencies greatly enhanced SWE and snow depth estimates, with mean normalized CRPS improved by $\tilde{1}0$ %. However, this
approach did not consistently improve results across all sites. Assimilating the frequency difference only slightly improved
snow depth estimates at TVC. These findings reflect the complex and sometimes contradictory results observed in previous
studies. Larue et al. (2018a, b) encountered mixed results when assimilating brightness temperatures at different frequencies,
with improvements in SWE RMSE when assimilating the difference of the frequencies in their synthetic experiment but the
opposite when using real data.

Rogers Pass, located in a alpine climate, presented unique challenges for data assimilation. Despite having the highest
number of synthetic backscatter observations (between 24 and 29), SWE estimates showed little improvement over the open
loop ensemble when assimilating backscatter. This seemed to be caused by backscatter signal saturation over a SWE threshold
of $\tilde{3}00$ kg m$^{-2}$ (Fig. 3). Backscatter simulated by SMRT fed by SVS2 snow outputs does not depend only on SWE, but also
on the vertical snow profile properties, particularly the SSA. This low threshold of SWE above which backscatter saturates can
be caused by several factors: 1) the SSA in SVS2/Crocus might be under-estimated in the simulations, which can then led to
overestimated modeled volume scattering in SMRT (Woolley et al., 2025; Vionnet et al., 2025) and 2) SMRT needs further
testing and development to estimate backscatter from deep alpine snowpacks at the considered frequencies. Future studies will
investigate the SWE threshold at which backscatter saturates based on SSA profiles. In addition, this experiment should be
conducted with cross-polarization backscatter that could have a stronger response to SWE than co-polarization (Borah et al.,
2022). Ongoing radar tower-based field experiments tend to indicate that the radar signal, even at 17.25 GHz, can penetrate
the maximum SWE observed at Rogers Pass as shown by Madore et al. (2023) who were able to retrieve SWE values up to a
1000 kg m$^{-2}$ with radar measurements at 24 GHz.



The assimilation results from assimilating backscatter observations were compared to assimilating synthetic SWE observations. Assimilating SWE observations with uncertainties less than 10 % provided the best estimates of SWE and snow depth at all sites; however, such small uncertainties are only realistic when observations are collected in-situ (e.g. Beaudoin-Galaise and Jutras, 2022). From SWE retrieval algorithms, larger uncertainties can be expected, and can be particularly high with large forest cover fractions (Foster et al., 2005; Cho et al., 2020; Pflug et al., 2024). When assimilating SWE with uncertainties below 20 %, the results in terms of bulk SWE and snow depth estimates were better than when assimilating the backscatter observations. However, the results were similar when assimilating SWE observations with an uncertainty of 30 %. This decrease in the quality of SWE estimates as SWE observation uncertainty increases is consistent with Shrestha and Barros (2025b), who also looked at SWE retrieval uncertainties between 5 % and 30 %.

Our study confirms the finding of Shrestha and Barros (2025b) and Phan et al. (2014) who showed improved vertical profile of density and SSA estimates after assimilation. Assimilating backscatter better improves snow density profiles than the SSA profiles. The distributions of normalized CRPS for the SSA profiles showed medians exceeding the mean, reflecting a small number of outliers with poor performance skewing the distributions. The normalized CRPS of density and SSA profiles were more improved at TVC than at the other sites, potentially because the Arctic version of SVS2 was used at that site, which impacts SSA estimates in the upper part of the snowpack (Vionnet et al., 2025). At the other sites, SSA was only improved when assimilating the SWE observations or the difference of backscatter at Rogers Pass (positive median). This improvement of SSA at TVC is promising as current multi-layered snowpack models struggle with representing Arctic snowpack stratigraphy (e.g Woolley et al., 2025; Vionnet et al., 2025).

## 4.2  Performance of the Particle Filter Algorithm

Despite the poor performance of the particle filter for high SWE values, the particle filter algorithm proved to greatly reduce the ensemble spread composed of the background particles when SWE values at the observation times were below 200 kg m$^{-2}$ (Fig. S5 of the supplementary material). Noticeable improvements of mean CRPS often above 40 % over the background ensemble were found for SWE values below 50 kg m$^{-2}$ at Powassan and Rogers Pass. This translated in the assimilation algorithm performing best during the early accumulation period and the melting period (Fig. 5). Similarly, Revuelto et al. (2021) showed that assimilating MODIS-reflectance with the particle filter performed better for shallower snowpacks. This highlights the value of assimilating observations early in the winter season, during the beginning of the snowpack accumulation phase. Early assimilation helps establish more accurate initial conditions, which in turn improves the effectiveness of assimilation throughout the season, especially during mid-winter.

Assimilation with the particle filter is inherently prone to degeneracy. Figure S6 of the supplementary material illustrates this, presenting the effective sample sizes ($N_{eff}$, Eq.7) averaged over 10 reference runs across different assimilation time steps, experiments, and sites. We observed that the algorithm typically degenerated either early in the snow season or towards the end of the melt season, with $N_{eff}$ values dropping below the 7 % threshold (Sect. 2.3.3). These periods of low $N_{eff}$ largely coincided with ephemeral snow events occurring before or after the main seasonal snowpack (Fig.s 2, S1, and S2). To mitigate degeneracy, systematic resampling was employed within the PF-SIR, and new perturbations were applied after each





assimilation time step. This strategy allowed the algorithm to successfully recover by propagating new particles with increased ensemble spread.

Overall, the particle filter was not prone to degeneracy during the middle of the snow season, with $N_{eff}$ values usually above 70 %, signifying good particle diversity and greater consistency between observations and ensemble predictions. In contrast, $N_{eff}$ values were lower during accumulation and ablation periods, when snow processes are most sensitive to meteorological forcing uncertainties (Günther et al., 2019) and model-observation mismatches are more pronounced. On average, assimilating the individual backscatter observations yielded slightly higher $N_{eff}$ values (+3 to +6) than assimilating SWE with 10 %
uncertainty, indicating that SWE observations were more discriminating and informative for constraining the particle ensemble.

### 4.3    Current Limitations

The assimilation method used in this study presents some limitations. The parameterizations of the different snow processes simulated by SVS2/Crocus were kept the same when generating the different members of the open loop ensembles, the reference runs, and the ensembles in the assimilation experiments. This was based on the assumption that meteorological forcings
are the main source of uncertainty in snowpack modelling (Günther et al., 2019; Raleigh et al., 2015). Future studies will consider an ensemble that accounts simultaneously for the uncertainties in the meteorological forcing as well as the snowpack model as done in Cluzet et al. (2021); Deschamps-Berger et al. (2022); this could improve the assimilation of backscatter observations at the Alpine site and reduce the backscatter saturation with observed SWE by having potentially more realistic SSA vertical profiles. The assimilation experiments were limited to 100 members as it has been found to be suitable for snow
assimilation experiments with the particle filter (e.g. Piazzi et al. (2018)). Although some studies have shown the possibility to apply other assimilation methods with multi-layered snowpack models, such as the Ensemble Kalman Filter and the 1D-Var (Shrestha and Barros, 2025b; Phan et al., 2014), they were not considered in our study.

### 5    Conclusions

This study investigates the potential of assimilating backscatter observations at two frequencies (13.5 GHz and 17.25 GHz) in
preparation for the TSMM satellite mission (Derksen et al., 2021). The synthetic assimilation experiments were conducted at three different sites spanning different Canadian climates, including an Arctic site (TVC), a humid continental site (Powassan), and an Alpine site (Rogers Pass). To test the assimilation for different snowpack conditions within each climate regime, three winter seasons at each site were considered, each with 10 random reference runs, which are assumed to represent the true snowpack state, from which the synthetic observations were extracted at a weekly interval (proposed measurement frequency
of TSMM) when the snowpack was dry. The results of the synthetic experiments are as follows:

- Assimilating individual backscatter improves SWE and snow depth estimates at all sites, with mean normalized CRPS against the open loop of SWE estimates up to 32 % at TVC, up to 23 % at Powassan, and up to 3 % at Rogers Pass.



- Assimilating both frequencies at the same time performed better at Powassan and TVC with a mean normalized CRPS of SWE estimates improved by ~10 % compared to the assimilation of individual frequencies.

- Assimilating the difference of the frequencies at the same time performed better at Rogers Pass with a mean normalized CRPS of SWE estimates over the open loop of 5 %.

- The SWE estimates obtained with backscatter assimilation were on par with estimates from assimilating SWE observations with high uncertainties (> 20 %), which can be expected from radar-based SWE retrievals (e.g., Pomerleau et al., 2020; Shrestha and Barros, 2025a).

- This study also showed that assimilating backscatter can improve the estimates of snow vertical profile properties, such as density and SSA, in some cases by up to ~10 % in terms of improved mean CRPS over the open loop.

This study provides initial insights into the assimilation of backscatter data directly within a snowpack model to improve predictions of SWE, snow depth, and vertical snow properties. The coupling of SVS2/Crocus and SMRT within the MuSA framework facilitates experimentation and advancements in snow data assimilation. Consequently, our work establishes the
foundation for an assimilation scheme tailored to future Ku-band SAR missions. A major limitation found in this study is the saturation of backscatter for SWE values above 300 kg m$^{-2}$, resulting in the limited improvements in SWE and snow depth estimates shown at the Alpine site. Work is currently done to overcome this limitation, particularly to improve the parameterization of SSA in SVS2 as well as to further test SMRT against field observations.

In the last few years, field studies have been conducted to gather data in preparation of TSMM. Kelly et al. (2024) obtained
measurements of backscatter at 13.5 GHz over the Powassan site during the 2022-2023 winter with the CryoSAR in conjunction with in-situ snow measurements of bulk snow properties and vertical snow properties. Similar data are available at TVC (Montpetit et al., 2025). The next steps of this study will include using these field data to test the methodology with real data at point-scale, comparing the performance of assimilating direct backscatter against retrieved SWE following the method of Montpetit et al. (2025), running an OSSE over a 2D domain using a simulator created to generate synthetic data using the
TSMM satellite orbit, and developing the 2D assimilation of backscatter within the Canadian Land Data Assimilation Scheme (CaLDAS) (Carrera et al., 2015).

*Code availability.* The implementation of SVS2-SMRT within the data assimilation platform MuSA is availabe in a permanent repository: https://10.5281/zenodo.17662807 (Leroux et al., 2025). The SVS2 code at point scale is available in a permanent repository: https://10.5281/zenodo.14859639 (Vionnet et al., 2025b). Finally, SMRT is also open-source and available on GitHub at
https://github.com/smrt-model/smrt.

*Author contributions.* NL led the integration of SVS2-SMRT within MuSA. NL and VV designed the overall experiment. CB contributed to generation of the open loop ensembles and JM and BM helped with the parameterization of SMRT. AD and MB helped with the design of



the assimilation experiments. NL did the simulations and the analysis of the results. MC, BB, MA, and FL provided guidance during all the steps of the study. NL drafted the manuscript and all authors participated in reviewing and editing the paper.

*Competing interests.* CD is a member of the editorial board of TC.



## Appendix A: Parameterization Used for the Crocus Simulations

**Table A1.** Parameterization used for the Crocus simulations. See Lafaysse et al. (2017), Vionnet et al. (2012), and Woolley et al. (2024) for a description of the parameterizations.

| Snow process | Default (Powassan and Rogers Pass) | Arctic (TVC) |
|---|---|---|
| Falling Snow Density | V12 | R21 |
| Snowdrift | VI13 | R21W |
| Snow Compaction | B92 | R2V |
| Thermal conductivity | Y81 | C11 |
| Radiative transfer | B92 | B92 |
| Liquid water transport | B92 | B02 |
| Metamorphism | B21 | B21 |



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
