# Peer review of "Assimilation of synthetic radar backscatters at Ku-band improves SWE estimates"

_EGUsphere, 2025_

## Referee Comment (RC2)

**General comments**

The authors present results from an experiment to assimilate modeled SWE (via SVS2/Crocus) and radar backscatter (via SMRT) into a timeseries of predictions of SWE and snow depth. The modeling framework was developed to support the Terrestrial Snow Mass Mission currently under development. Overall this work is relevant for the snow hydrology and remote sensing communities and is appropriate for The Cryosphere.

The manuscript would benefit from more precise language throughout. Specifically, I think it is important to be explicit that no snow or radar observations are actually used in this analysis; everything is a model output. The performance statistics are derived using 10 reference runs which are model output generated with perturbed meteorological data. Given this setup, I do not think it's appropriate to refer to the input data as an "observation". This is fairly inconsistent throughout the paper, e.g. "backscatter observations" (line 268) and "SWE observations" (line 269), but sometimes there are qualifiers like "synthetic SWE observations" (line 276) or "synthetic true states providing reference snow states and backscatter values" (line 182). I suggest using phrases like modeled SWE, simulated backscatter, synthetic data, etc. and being consistent throughout the paper. I think the use of "reference" to describe a model run used as the basis for comparison is effective. Maybe it's possible to use that terminology to refer to variables as well.

My other general comment is about the ensemble generation, specifically the assumption in lines 171-172 that the HRDPS model errors are consistent between the three sites. This assumption could use some additional justification because the three sites are described as having very different characteristics in Section 2.1. In addition to the Powassan site having warmer temperatures and much less snow than Rogers Pass, I assume the rolling terrain around Powassan is much better represented in the 2.5 km HRDPS model than the complex terrain around Rogers Pass. So I am not convinced that the HRDPS errors would necessarily be of similar magnitudes at these sites, especially for precipitation which you state can be a significant error source for SWE estimates in line 28. I think the perturbation distributions could actually be much wider at Rogers Pass, which would in turn impact most/all of the subsequent results. If you have long-term timeseries of both snow depth and SWE at both sites, one way to check (not the only way!) would be to compare the ensemble spread shown in rows 3 and 4 of Figure 2 to many seasons of observations at the three sites. If a similar percentage of seasonal snow depth/SWE curves fall within the spread of the model at all sites, that could be evidence that the method is fine. If fewer seasons fall within the spread at some sites (e.g. Rogers Pass compared to Powassan), you may need to widen the perturbation distributions for those sites. I'm not sure the best way to perform this analysis quantitatively to look for significant differences between sites, but maybe you can somehow start with the distributions in Table 2.

**Specific comments**
Title: The phrase "Assimilation of synthetic radar backscatters" in the title is a bit unclear, given that TSMM is a synthetic aperture radar mission but here "synthetic radar" is used to refer to the output of the SMRT model. I suggest replacing "synthetic" in the title to simulated or modeled (or another synonym of your choosing).

Line 69: "if retrieval algorithms are able to retrieve a wider range of SWE values" – does this imply the need for backscatter measurements to be obtained over a wide range of snow conditions, or an appropriate algorithm formulation that can produce a large range of SWE values from the backscatter data? Put another way, is this in reference to the backscatter data or the algorithm?

Line 145: Please be more specific with the forecasts and lead times here. For example, with the 7-12 hour lead time, are HRDPS forecasts generated only every 12 hours but with hourly lead times that allow you to fill in the hourly timeseries? Or are there hourly forecasts that allow you to build your timeseries always with 1 hour lead times?

Line 150-156: I think I understand that this perturbation method is applied to multiple meteorological variables. I suggest making this more explicit in the text here, either using another subscript in Equation 1 (though this may end up looking too messy) or a sentence before Equation 1. If there are multiple variables considered, do they all have the same decorrelation time length (tau)? How is tau calculated?

Line 157: How did you select additive vs. multiplicative for a given meteorological variable?

Line 169-170: Okay, this info helps answer my previous question. I suggest moving these lines (and perhaps Table 2) closer to Equation 1 in the text. Especially the detail about calculating alpha as the autocorrelation of HRDPS residuals – that part is not clear from the initial presentation of the equations.

Figure 2: I suggest making the linewidths of the red reference runs a little narrower so it's easier to compare them to the gray shading of the ensemble spread.

Line 275: "despite this site having the highest number of weekly observations throughout the winter" – I thought all sites had the same weekly data (12 UTC on Mondays pulled from the hourly data to run the model). Do the assimilation timesteps vary between sites? I also notice a dashed line for "Obs. Times" in the Figure 3 legend but I don't see any of those lines in the panels.

Line 312-314: Is it possible that the improvement seen during the melt season is influenced by the sample size? I imagine the modeled LWC reaches 1% some days/weeks before the SWE actually starts decreasing. In that case, how many backscatter simulations actually get assimilated during the melt season at these sites, especially in comparison to the earlier periods? I see in Figure 5 that the latest period boxplots at all sites have higher mean/median CRPS than the earlier periods, but they also have much wider spread. So are they statistically significantly higher in the latest periods? It would be helpful to run quantitative significance tests on these results, and also include sample sizes in the x tick labels, e.g. "before Dec (n=XXX)".

Figure 5: It would be helpful to make this figure larger to see more detail. Consider a 2x3 subpanel layout (snow depth and SWE columns with sites as different rows) instead of 3x2.

Figure 5: It is probably expected that assimilating SWE gives better results for both SWE and snow depth when the assimilated SWE errors are smaller. But it is interesting that the spread of the SWE assimilations seems to decrease with increasing error (i.e. taller boxplots for SWE with 5% error than 30% error). This looks like the opposite trend compared to Figure 4. Can you provide more discussion around these results?

Figure 7: It would be helpful here as well to have an indication of the sample sizes going into these boxplots, as well as some statistical interpretation of the results. Is it one vertical profile every hour for all 10 reference runs, or only profiles from December 27 1200 UTC every year? Given that the interquartile ranges of almost all boxplots in Figure 7 span both positive and negative values, are the results significantly different from a CRPS of 0? Put another way, does assimilating backscatter or SWE actually improve the modeled density of SSA?

Line 344-350: This paragraph seems better suited for the conclusion of the paper.

Line 361-362: "Consequently, the higher number of assimilated observations at TVC may have contributed to the improved SWE and snow depth estimates." – this is contrary to your previous observation that Rogers Pass had the worst performance with the most assimilated data points (line 275, also 377).

Line 393: Consider adding Bonnell et al. (2024) to the reference list here. They use an InSAR approach with much lower frequency but I think it's an important point that dense forests are going to be tricky for any SAR-based approach for SWE retrievals. (Full disclosure – I am on this paper.)

Line 398-401: I'm not convinced of these statements based on the results of Figure 7 (see above comment). Please revisit after some more rigorous analysis of the density and SSA results.

Line 402: Pointing to the different SVS2 setup is unsatisfying here because the whole modeling experiment is predicated on errors being from the meteorological forcing data, not the model (lines 141-142).

Section 4.2: This is an interesting point that could use more discussion. The assimilation resulted in the best improvement during the early season, which is exactly when we would expect ephemeral snow events that lead to the algorithm degeneracy. Figure S6 is quite noisy but it looks like the backscatter sample sizes show bigger drops in the early season compared to SWE sample sizes. Is this because snow melts and increases the soil moisture, which drops the modeled backscatter below the acceptable threshold? In this experiment it's possible to generate some new particles, but what about the approach with actual satellite data?

**Technical**
Line 3: capitalize Change
Line 16: ~ appears over the 3 instead of before it. Maybe a \sim instead of a \tilde? A few other occurrences of this throughout the manuscript.
Line 68: "RMSEs were" or "RMSE was"
Lines 94-97: Check for consistency between Section and Sect.
Line 101: see → sea
Line 105: Montpetit et al. missing a year
Line 116: remove "detailed"
Line 117: 1D → 1-dimensional (for clarity)
Line 135: maybe "the version 2.0 of the MuSA" → "MuSA version 2.0"
Line 142: change final comma to period.
Figure 2 axes and caption: snow height → snow depth (for consistency with the rest of the paper)
Line 281: maybe "either backscatter" → "backscatter at either frequency"

**References**

Bonnell, R., Elder, K., McGrath, D., Marshall, H. P., Starr, B., Adebisi, N., Palomaki, R. T., and Hoppinen, Z.: L-band InSAR snow water equivalent retrieval uncertainty increases with forest cover fraction, Geophysical Research Letters, 51, e2024GL111708, https://doi.org/10.1029/2024GL111708, 2024.